# Electrical Potentiometry with Intraoral Applications

**DOI:** 10.3390/ma15155100

**Published:** 2022-07-22

**Authors:** Alfonso Jornet-García, Arturo Sanchez-Perez, José María Montoya-Carralero, María José Moya-Villaescusa

**Affiliations:** Department of Periodontology, Medicine and Dentistry Faculty, Murcia University, 30008 Murcia, Spain; alfonsofelipe.jornet@um.es (A.J.-G.).; jmmontoya@um.es (J.M.M.-C.); mjm.villaescusa@um.es (M.J.M.-V.)

**Keywords:** corrosion, dental implants, biocompatible materials/chemistry, electrochemical techniques, saliva, artificial/chemistry

## Abstract

Dental implants currently in use are mainly made of titanium or titanium alloys. As these metallic elements are immersed in an electrolytic medium, galvanic currents are produced between them or with other metals present in the mouth. These bimetallic currents have three potentially harmful effects on the patient: micro-discharges, corrosion, and finally, the dispersion of metal ions or their oxides, all of which have been extensively demonstrated in vitro. In this original work, a system for measuring the potentials generated in vivo is developed. Specifically, it is an electrogalvanic measurements system coupled with a periodontal probe that allows measurement of the potentials in the peri-implant sulcus. This device was tested and verified in vitro to guarantee its applicability in vivo. As a conclusion, this system is able to detect galvanic currents in vitro and it can be considered capable of being employed in vivo, so to assess the effects they may cause on dental implants.

## 1. Introduction

“Galvanism” is named after the Italian physician Luigi Galvani (1737–1798) (Gillispie) [1], who was the first to describe the effects of electrical currents on living tissues. In dentistry, the term “galvanism” is used to describe the electrical currents that are established between metals of different characteristics.

Galvanism has long been associated with various oral manifestations, such as pain or metallic taste [2], leukoplakia [3], lichen planus [4], and toxic or allergic reactions to corrosion products [5]. Recently, it was also identified as a possible cause of implant loss [6].

These currents are established when two or more different metals with different electrical potentials are placed in the oral medium, forming an electrochemical cell. In this way, oxidation on the surface of one metal (the anode) and reduction on that of the other (the cathode) occur. This exchange occurs through saliva, which acts as an electrolyte. Both metals then interact, as shown in Figure 1.

For more than 70 years, commercial pure titanium implants (Ti implants) have been used to replace missing teeth [7]. Dental implants have been shown to be a reliable option for tooth loss with high survival and success rates [8]. Therefore, their use has become widespread, although they are not without complications [9]. These complications can lead to implant loss, which is usually attributed to both biological and mechanical factors and a combination of local and systemic factors [9,10].

Commercial pure titanium (cp-Ti) and titanium alloys (typically Ti6Al4V) display excellent corrosion resistance and biocompatibility. Although zirconium implants are available, this study focused on titanium and its alloys because these are the most frequently used implants.

Several studies have shown that Ti implants inserted into bone can continuously release Ti particles [11,12]. This degradation of the Ti implant depends on the oral environment, which can be highly corrosive in different ways (Figure 2), affecting the behaviour of the implant and its prosthesis and releasing Ti particles [13,14,15,16,17].

These Ti particles can enhance the expression of inflammatory cytokines by activating osteoclasts, macrophages, and neutrophils [14,15,18,19,20].

Most studies on the effect of corrosion have been performed in vitro due to the difficulty in developing a measurement system that can be used intraorally [21,22].

The purpose of this study was to develop a system for measuring galvanic currents that would allow us to perform accurate measurements in the oral cavity. On the other hand, we also sought to compare the new device with a reference electrode.

## 2. Materials and Methods

### 2.1. Materials

An HI 9025 pH meter (Hanna Instruments SL, Eibar, Municipality of Eibar, Spain) 07was used to measure the electrical potentials on the surface of the implants (Figure 3). Its main characteristics can be found at the following link: http://www.ictsl.net/productos/aparatos/phmetroportatilhi9024yhi9025hanna.html (accessed on 7 June 2022).

For the measuring electrode, an original “periodontal electrode” was designed. This electrode consists of a reference electrode (model ME 402; Radiometer Analytical SA, France) (Figure 4a) connected to a stainless steel periodontal probe (periodontal probe #8 handle #6 Qulix 3-6-8-11; Hu-Friedy) (Figure 4b) that acts as a contact electrode. To validate this electrode, it was compared with an oxidation–reduction potential (ORP) combined electrode with a platinum tip and an integrated reference electrode (model HI 3131P; Hanna Instruments SL, Eibar, Spain) (Figure 4c).

All measurements were made at 37 ± 0.5 °C in a Precisterm Selecta Model 6,000,137 thermostatic bath (Laboquimia, Lardero, La Rioja, Spain). The temperature was checked with an HI 7669/2W temperature probe (Laboquimia, Lardero, La Rioja, Spain) and an HI 9025 pH metre (Laboquimia, Lardero, La Rioja, Spain).

To simulate oral conditions, an artificial saliva bath was used according to the formula shown in Table 1.

### 2.2. Method

The potentiometry measurements were performed in and validated by a simulation in a bath with artificial saliva. In this way, the range of possible values was evaluated in vitro and the measurement method was fine-tuned.

A total of 50 measurements (25 with each electrode) were made on 25 samples of titanium implants, each 12 mm long and 3.75 mm wide (Microdent System, Barcelona, Spain).

The electrodes were maintained using a series of storage, calibration, and disinfection solutions.

The ME 402 reference electrode was stored upright in a saturated solution of KCl (potassium chloride), ref. S21M010 (Radiometer Analytical SA, Villeurbanne Cedex, France), and the vent hole was protected from the environment. Before each potentiometry measurement, the electrodes and the measuring instrument were checked with a 468 mV (25 °C) Redox Pattern solution (Crison, code 9410, Barcelona, Spain). Before and after each measurement, we cleaned the electrochemical probes with RENOVO.N (S16M001; Hach Lange, Berlin, Germany) electrode cleaning solution (potassium hydroxide).

To calibrate the pH scale of the pH metre, we used two buffer solutions, both from Crison (Barcelona, Spain): one at pH 4.01 (25 °C; code 9463) and one at pH 7.00 (25 °C; code 9464).

Finally, Terg-A-Zyme, an enzymatic detergent (1304; Alconox, Inc., New York, NY, USA) was used to clean all the instruments once the in vitro study was completed and before and after each clinical measurement.

To validate the measurement method, the measurements were repeated 25 times by the same observer under identical conditions; measurements with absolute difference from the mean of more than 5% were considered poor and rejected.

### 2.3. Statistical Analysis

Statistical analysis of the data collected in this study was carried out using the statistical package SPSS (Statistical Package for Social Sciences), version 25.0 (IBM Corp., Armonk, NY, USA). This software was also used to perform the following procedures: descriptive statistics, frequency analysis, and Student’s *t*-test. *p* < 0.05 was accepted as statistically significant.

## 3. Results

A total of 50 measurements were performed (25 with each electrode) for 25 samples of Microdent System titanium implants measuring 12 mm in length and 3.75 mm in width. These measurements were performed in artificial saliva at 37 ± 0.5 °C, yielding the following results:

Both the combined electrode and the periodontal electrode recorded an absolute potential value of approximately 243.26 mV (CI 95% 232.43–254.09). The mean for the combined electrode was 210.28 mV (CI 95%205.09–215.47), and that for the periodontal electrode was 276.24 mV (CI 95% 266.64–285.84). Table 2 presents the descriptive analysis of the electrical potential value for the combined and periodontal electrode in terms of central tendency, dispersion, position, asymmetry, and kurtosis (Table 2)

The expected values were normal in both cases and demonstrated similar normal Q–Q plots (Figure 5).

The box plots of the electrodes show a wider dispersion for the periodontic electrode potential, but the values are higher than those of the combined electrode (Figure 6).

These differences were statistically significant according to the paired Student’s t-test (Table 3).

It is worth mentioning that the relative values were negative for the combined electrode and positive for the periodontal electrode, so the analysis was performed with the absolute values.

## 4. Discussion

Titanium (Ti) is characterized by its resistance to corrosion because it forms an oxide layer that protects its interior. This phenomenon is called “passivation” [23].

Despite the presence of this outer layer, there is some evidence that Ti implants can release Ti particles slowly but continuously [24]. The importance of these particles seems to be key to the local response of the organism [25,26,27]. Previous studies have verified a concentration gradient of particles in the tissue, with higher concentrations observed closer to the implant [24,28,29].

The progressive accumulation of these particles, due to their low solubility, can cause a response that triggers the loss of bone around the implant and finally, the loss of the implant itself [30]. The mechanism by which this local accumulation of particles can trigger the onset and progression of peri-implantitis continues to be unclear. As previously described, reactive oxygen species (ROS) generation could be responsible for the recruitment of neutrophils. However, the physiological purpose of this oxidative regulatory mechanism remains unknown [31]. Ultimately, this process results in an imbalance between osteoblasts and osteoclasts [32].

As mentioned above, although the highest concentration is local to the implant, the effects of Ti particles can also be seen at the systemic level [11,12]. Small concentrations have been found in distant organs, such as the lung, liver, spleen, and kidneys [12]. However, in general, they do not produce a systemic immune response [24].

One of the causes potentially influencing the release of Ti particles is corrosion due to the formation of electrical potentials, which can determine their local accumulation [33,34,35] (Figure 7).

The system was able to quantify the electrical potentials that can be generated around dental implants through the use of Evans diagrams. In this way, it was possible to determine both the intensity of the electric currents and the rate of the corrosive reactions. It can be considered that these measurements can be useful for reducing corrosion and preventing the appearance of subsequent complications.

The measurement system is similar to those used by other researchers, such as Sutow et al. and Muller et al. [21,22], to record the electrical potentials of metal restorations. However, the adaptation of a periodontal probe provides us with clinical advantages such as the ability to record electrical potentials within the gingival sulcus.

This new method opens numerous avenues of study, including identification of the processes that develop inside the periodontal groove of restored teeth, quantification of the electrical activity that occurs in the peri-implant groove, and assessment of the relationship between oral electrical potentials and oral diseases attributed to galvanism (pain, oral lichen, and leukoplakia, among others) [2,3,4,5].

Multiple factors can facilitate the release of these particles, including the surface treatment of the implant, friction during implant insertion, corrosion of the implant surface, friction at the implant–abutment interface, the tissue microenvironment, implantoplasty, and the method used for implant surface detoxification [30,36]. Among these, corrosion and friction wear have been mentioned as particularly important causes [37,38].

The effect of corrosion is considered inevitable for almost all existing metal implants [39]. As mentioned above, this release will last for years [40]. However, its intensity is influenced by the oral environment, in which both the temperature and the pH of the saliva play a prominent role, and even by certain bacteria that are capable of eroding the titanium layer, releasing Ti particles [6].

To understand this effect, it should be considered that a dental implant is submerged in saliva, favouring the creation of a galvanic cell and therefore causing an electrolytic effect on the surface of the implant [33]. This effect facilitates the release of Ti particles in the surrounding tissues [34,35]. In extreme situations, when this environment is maintained for years, a loss of strength can occur in the implant core [41], which can lead to a catastrophic failure, culminating with fracture of the implant itself [42].

As mentioned above, any factor that alters the saliva will have a potential effect on implant corrosion and release of Ti particles [43,44]. Therefore, our first objective was to verify the feasibility of the procedure in a controlled environment.

To establish the usefulness of the proposed system, a medium was used that reproduces artificial saliva and allows us to control the pH and temperature. The results support the use of this method; we recorded electrical potentials with the periodontal electrode similar to those of other authors, with a mean value of 276.24 mV (CI 215.47–285.84). These values can be greater in polymetallic conditions, that is, when more than one metal is present in the oral cavity, such as with old amalgam fillings, superstructures or abutments of different compositions [45].

Another variable to consider is the composition of Ti implants manufactured with titanium alloys. Specifically, Cai et al. measured the in vitro open circuit potential (OCP) of four different titanium alloys with three different surface treatments. Their results show that the electrical potential of uncoupled titanium alloys varied from 85 mV to 205 mV [46].

The results agree with the ones of that study, although the influence of temperature seems to be of such importance that it should be taken into account when determining the potential. Other authors even determined that metallic combinations may pose a greater risk, so they established a series of clinical guidelines for the use of combination metallic restorations [47].

Among the immediate applications of this measurement method are the in vitro study of the factors that can modify the electrical potential of titanium implants (such as the surface area, implant length and diameter, restorative material, medium or temperature variations) and the in vivo study of the electrical potential of dental implants subjected to load, as well as the relationship between this potential and variables such as bone loss or peri-implant inflammation. In conclusion, this system is capable of measurements in the 250Mvolt range and can be used in vivo on patients with implants. This will allow to study the possible effect of corrosion on bone loss around implants.

### Limitations

The measurement of electrical potentials includes errors that are inherent to the electrical circuit. One of the main sources of error is the drop in IR of the measurement circuit, where I is the current and R is the total resistance. The total resistance of the circuit includes the resistance of the voltmeter, all contact resistances, the resistance of the electrolytes, and the resistance of the tissues between the reference electrode and the measurement electrode [48]. However, despite all the errors inherent to the circuit, the clinical measurement of the electrical potential of metal restorations showed high reproducibility: differences of only 2 mV were obtained across three independent measurements at intervals of 5 s [20].

It should also be noted that the behavior of the galvanic currents will depend on the material used (Soares and Stich) [49,50], the treatment of peri-implantitis (Lozano and Verdeguer) [51,52], as well as the development of future materials; therefore, the study of Ti particles and their effect must be taken into account (Gürbüz-Urvasızoğlu G) [53] at least until new particle-free materials become available (Nagay) [54].

## Figures and Tables

**Figure 1 materials-15-05100-f001:**
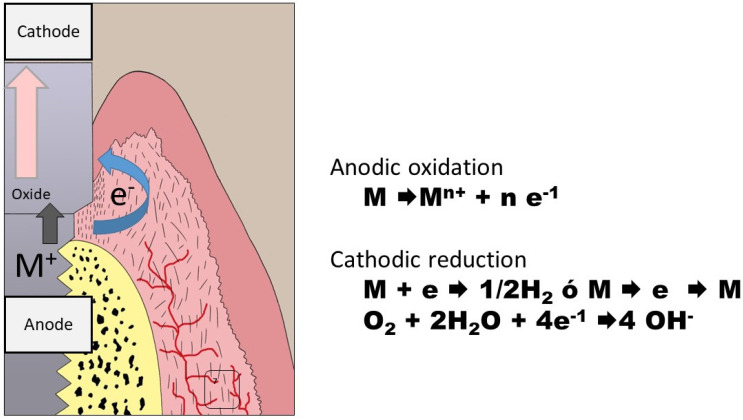
The anode, the metal that is oxidized (M), releases ions (M^n+^) to the medium and an equal number of electrons (n) as the valence of these ions (M^n+^). The metal that is reduced can incorporate these electrons (e) both on its surface and in its metal lattice by reacting with both hydrogen and oxygen, depending on the medium.

**Figure 2 materials-15-05100-f002:**
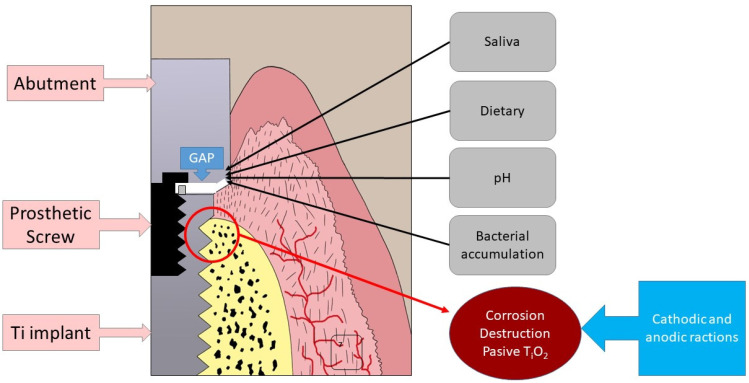
Different ways in which a dental implant can corrode and lose its passivated layer.

**Figure 3 materials-15-05100-f003:**
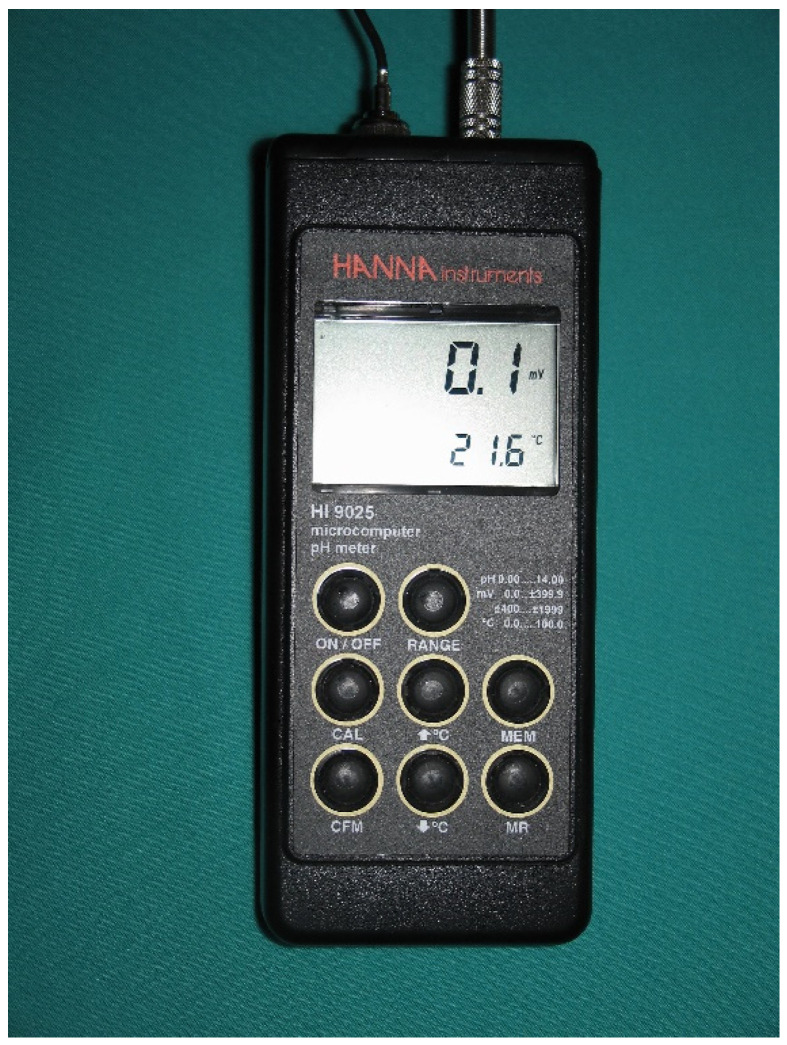
The Hanna waterproof instrument, a heavy-duty pH meter designed to provide accurate laboratory results.

**Figure 4 materials-15-05100-f004:**
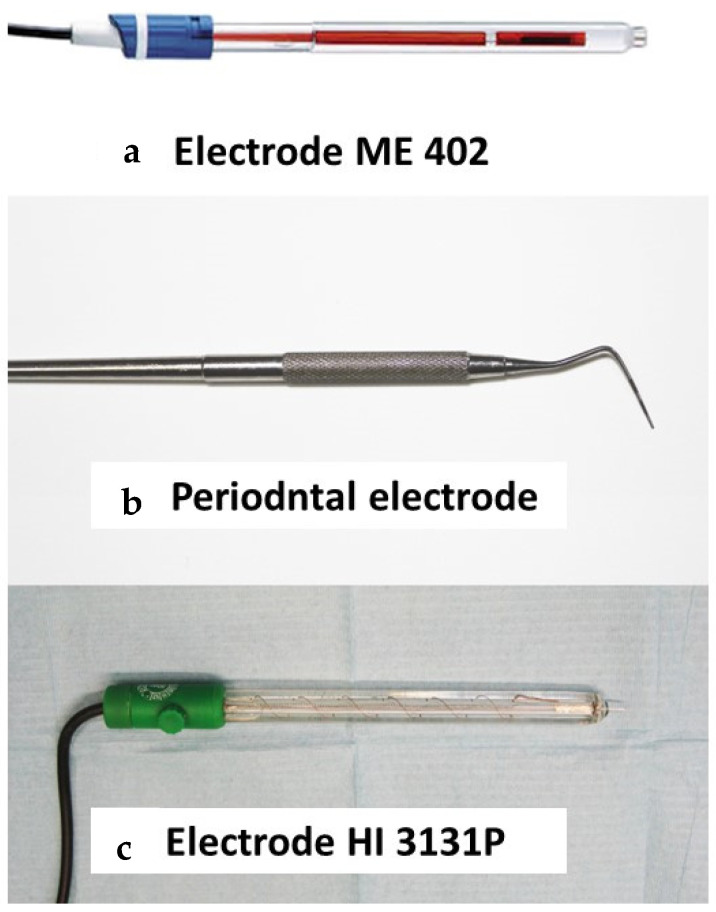
Different electrodes used in these measurements. (**a**) Reference electrode suitable for general laboratory use. (**b**) Stainless steel periodontal probe. (**c**) Combined electrode with platinum tip and integrated reference electrode.

**Figure 5 materials-15-05100-f005:**
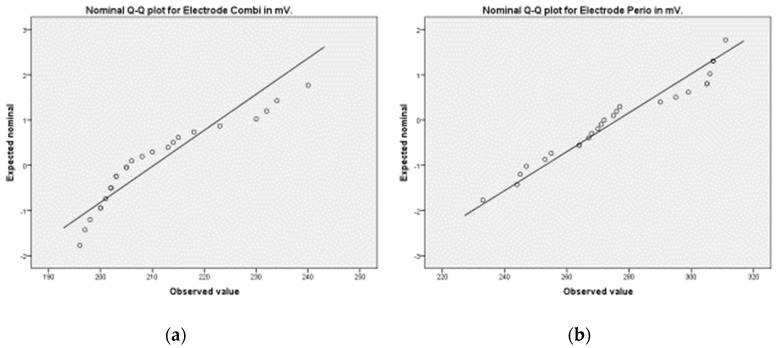
Statistics, Q-Q(quantile-quantile) plots play a ital role to graphically analyze and compare two probability distributions Normal Q–Q plot. (**a**) Normal distribution of the combination electrode potential in mV. (**b**) Normal distribution of the periodontic electrode potential in mV.

**Figure 6 materials-15-05100-f006:**
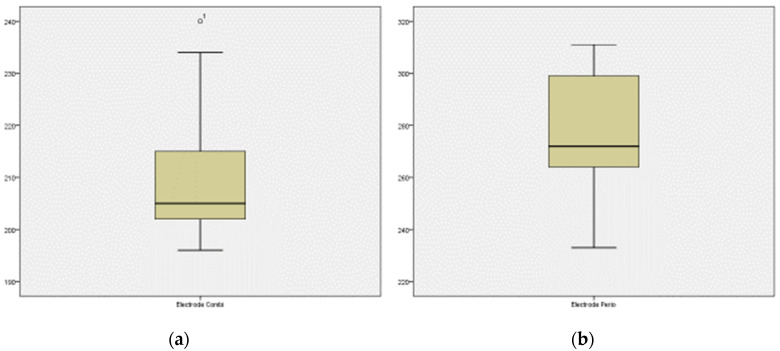
Box plot of the total bone-to-implant contact (BIC) obtained with control and UVC-treated surfaces. Plots show the lower (Q1), median (Q2), and upper (Q3) quartiles. The whiskers indicate the highest and lowest observations. (**a**) Box plot of the combination electrode potential in mV. (**b**) Box plot of the periodontal electrode potential in mV.

**Figure 7 materials-15-05100-f007:**
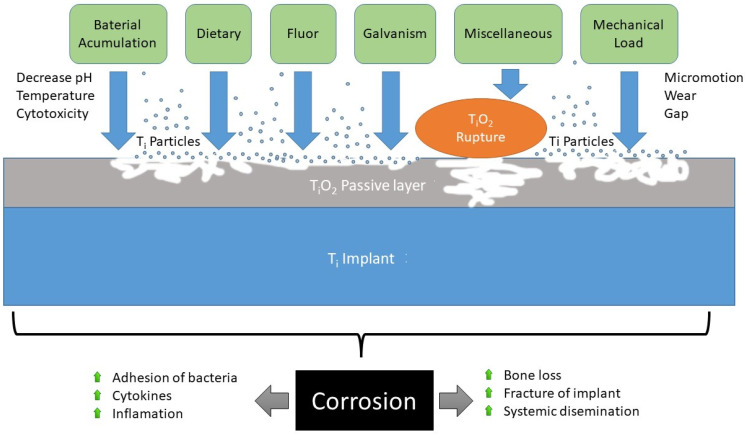
Diagram of the different mechanisms capable of causing Ti particle release and rupturing the passivated titanium layer.

**Table 1 materials-15-05100-t001:** Composition of the artificial saliva used to conduct the experiments.

Chemical Components of Artificial Saliva		
Sodium Biphosphate 200 mg	Potassium Chloride 1.2 gr	Potassium Thiocyanate 330 mg
Sodium Bisphosphate 260 mg	Sodium Chloride 700 mg	Sodium Bicarbonate 1.5 gr
Urea 1.5 gr	Purified water sqt 1000 ml	Lactic acid sqt pH 6–7

mg = milligrams = grams, ml = millilitres, pH = potential of hydrogen, sqt = “sufficient quantity to”.

**Table 2 materials-15-05100-t002:** Descriptive analysis of the electrodes. Results in mV.

Statistical	Combined Electrode	Periodontal Electrode
Mean	276.24	−210.28
Lower limit of the CI *	266.64	−215.47
Upper limit of the CI *	285.84	−205.09
5% trimmed mean	276.63	−209.48
Median	272	−205
Variance	540.44	157.96
Standard deviation	23.24	12.56
Minimum	233	−240
Maximum	311	−196
Range	78	44
Interquartile range	43	15
Asymmetry	−0.00	−1.07
Kurtosis	−1.09	0.12

* 95% confidence interval for the mean.

**Table 3 materials-15-05100-t003:** *T*-test values. There were statistically significant differences between the two electrodes. The values recorded with the periodontal electrode were higher than those of the combined electrode.

*T*-Test	*T*-Test Value	*p*
Values	−189.99	0.000

## Data Availability

The data of this study are available to readers upon request to the corresponding author (arturosa@um.es) in Excel format.

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
