# Peer review of "Electrical Potentiometry with Intraoral Applications"

_materials, 2022, doi:10.3390/ma15155100_

Round 1

Reviewer 1 Report

An electrogalvanic measurement system coupled to a periodontal probe that allows measurement of the potentials in the peri-implant sulcus was developed in this work. It can be accepted for publication after the following issues are clarified.

1. As discussed in the review (Recent progress of novel biodegradable zinc alloys: from the perspective of strengthening and toughening, Journal of Materials Research and Technology, 2022, 17: 244-269), various metallic biomaterials have been developed for application as medical implants. Authors selected Ti alloys as study subject. It’s better to introduce the advantages of Ti alloys compared to other metallic biomaterials in the Introduction section.

2. The quality of Figures 5 and 6 should be improved.

3. Is the system developed in this work applicable to other metallic medical implants?

4. All tables should be replaced by standard three-line tables.

5. What are the advantages to the system developed in this work compared to others?

6. How to overcome the limitations of the system mentioned in this work?

Author Response

Referee 1

Comments and Suggestions for Authors

An electrogalvanic measurement system coupled to a periodontal probe that allows measurement of the potentials in the peri-implant sulcus was developed in this work. It can be accepted for publication after the following issues are clarified.

  1. As discussed in the review (Recent progress of novel biodegradable zinc alloys: from the perspective of strengthening and toughening, Journal of Materials Research and Technology, 2022, 17: 244-269), various metallic biomaterials have been developed for application as medical implants. Authors selected Ti alloys as study subject. It’s better to introduce the advantages of Ti alloys compared to other metallic biomaterials in the Introductionsection.

Answer:Although the chemical composition and topography are considered important, the mechanical properties of the material and the loading conditions in the host have, conventionally, influenced material selection for different clinical applications: predominantly Ti6Al4V in orthopaedics while cp-Ti in dentistry.

Earliest Titanium implants (in dentistry) were made of pure titanium. It is believed that pure titanium optimized osteoblastic differentiation much more than did Ti alloy. In the long term commercially pure Titanium implants were more stable and showed slightly better Bone Implant contact (BIC) as well as better Removal torque (RT).

Indeed, in recent times other materials have appeared for use in dental implantology, among them zirconium and PEEK (Poly Ether Ether Ketone), however, the implants that are usually used predominantly are made of titanium or titanium alloys. For this reason, we use titanium for the fine-tuning of our registration method.

We have included the following sentence in the introduction as suggested by the referee.

“Commercially pure titanium (cp-Ti) and titanium alloys (typically Ti6Al4V) display excellent corrosion resistance and biocompatibility. Although zirconium implants are available, our study focused on titanium and its alloys because these are the most frequently used implants.”

  1. The quality of Figures 5 and 6 should be improved.

Answer:We have taken photos 5 and 6 again in high resolution. Thank you

  1. Is the system developed in this work applicable to other metallic medical implants?

Answer:Since the measurement system is universal, it can be used to measure any metallic material capable of producing electrical currents. Our idea is that this design can be used intraorally within the sulcus that separates the gingiva from the implant.

In addition to this, it can be used in any situation where a galvanic pile between any 2 metals anywhere in the body can occur as long as they are immersed in a conductive liquid (serum, blood, saliva ...).

  1. All tables should be replaced by standard three-line tables.

Answer:We have redone the tables in a 3-line format

  1. What are the advantages to the system developed in this work compared to others?

Answer:Its main advantage is that it can be used inside the patient's mouth and enhance in vivo studies. Its small dimensions and its coupling to a periodontal probe allow us to carry out very localised measurements in the areas of interest.

  1. How to overcome the limitations of the system mentioned in this work?

Answer:Since the errors are systematic for a given measurement system, and not random, once determined they will be constant and for the same reason. By repeating themselves homogeneously, they will always mark a constant direction in the potentials. IR will remain a constant, reproducible and reliable.

Thank you for your help in improving the study

Reviewer 2 Report

The present manuscript summarize the results of an in vitro electro-galvanic measurement system coupled to a periodontal probe that allowing measurement of the potentials in the peri-implant area. This electro-galvanic measurement system allowed identification of the potentials generated between different components of the dental implant in an easy and economical manner, thus  performing effective  accurate measurements in the oral cavity.

Strength points:

-Originality/Novelty: The objective of the article is original and well defined.

-The introduction of the results provides sufficient background by including relevant references

-Interest to the Readers: The article has potential to attract a fairly large readership.

-Overall Merit: The article has potential to initiate an advance in the current knowledge by providing results of interest for both research and medical practice.

-English Level: The English language is appropriate and understandable.

Weakness points:

The article presents a quite limited approach of an initial in vitro testing without studying the influence  of various factors on the electrical potential of titanium implants (surface area, restorative material, medium or temperature variations). Also, in vivo  study of the electrical potential of dental implants subjected to load, as well as the relationship between this potential and variables such as bone loss or peri-implant inflammation are missing.

Author Response

Referee 2

Comments and Suggestions for Authors

The present manuscript summarize the results of an in vitro electro-galvanic measurement system coupled to a periodontal probe that allowing measurement of the potentials in the peri-implant area. This electro-galvanic measurement system allowed identification of the potentials generated between different components of the dental implant in an easy and economical manner, thus  performing effective  accurate measurements in the oral cavity.

Strength points:

-Originality/Novelty: The objective of the article is original and well defined.

-The introduction of the results provides sufficient background by including relevant references

-Interest to the Readers: The article has potential to attract a fairly large readership.

-Overall Merit: The article has potential to initiate an advance in the current knowledge by providing results of interest for both research and medical practice.

-English Level: The English language is appropriate and understandable.

Weakness points:

The article presents a quite limited approach of an initial in vitro testing without studying the influence of various factors on the electrical potential of titanium implants (surface area, restorative material, medium or temperature variations). Also, in vivo study of the electrical potential of dental implants subjected to load, as well as the relationship between this potential and variables such as bone loss or peri-implant inflammation are missing.

Answer:We totally agree with the referee. The objective of this work was to fine-tune our measurement method. Once achieved, our next objectives to be published are:

  • To determine the influence of implant length and diameter. As this implies an increase in the metallic core.
  • To determine the effect of temperature for a standard implant.
  • To determine the effect of pH for a standard implant.
  • Determine the effect of fluoride for a standard implant.
  • To determine in vivo if electric currents correlate with the presence of peri-implantitis.
  • To determine in vivo in humans if electric currents affect the bone level.

Answer:We hope to be able to develop the next phases of the research once our method is published.

Answer:We have included the following sentence at the end of the discussion. Page 9 lines 229-235:

"Among the immediate applications of our measurement method are the in vitro study of the factors that can modify the electrical potential of titanium implants (such as the surface area, implant length and diameter, restorative material, medium or temperature variations) and the in vivo study of the electrical potential of dental implants subjected to load, as well as the rela-tionship between this potential and variables such as bone loss or peri-implant inflam-mation.”

Thank you for your suggestions

Reviewer 3 Report

The paper shows results about electrical potentiometry with intraoral applications. Authors developed a system for measuring the potentials generated in vivo. They elaborated electrogalvanic measurement system coupled to a periodontal. It was tested and verified in vitro to guarantee its applicability in vivo. Mentioned system allows identification of the potentials generated between various components of the dental implant.

Dear authors. Thank you very much for your value paper about study of electrical potentiometry with intraoral application. This kind of study is well connection of various groups of science such as medicine, chemistry, electrical engineering. The results can find real applications. I have some comments and suggestions, what should be considered in presented to review paper.

Comments and suggestions:

1. Line 22 – “… Italian physician Luigi Galvani (1937-1798), who …”. I think, the years are not correct. Please correct.

2. Introduction chapter well describes fundamental information about galvanism and study presented in the paper. References are correct.

3. Is Figure 1 made by authors or taken from references. If from references, add information what reference it is.

4. Subchapter 2.1. Materials. I think, it would be good to present some technical data of used measurement device, such as minimum of voltage, resolution, and others. Please complete if it is possible.

5. Table 3 – please use the units of properties, you present in the table.

Author Response

Referee 3

Comments and Suggestions for Authors

The paper shows results about electrical potentiometry with intraoral applications. Authors developed a system for measuring the potentials generated in vivo. They elaborated electrogalvanic measurement system coupled to a periodontal. It was tested and verified in vitro to guarantee its applicability in vivo. Mentioned system allows identification of the potentials generated between various components of the dental implant.

Dear authors. Thank you very much for your value paper about study of electrical potentiometry with intraoral application. This kind of study is well connection of various groups of science such as medicine, chemistry, electrical engineering. The results can find real applications. I have some comments and suggestions, what should be considered in presented to review paper.

Comments and suggestions:

  1. Line 22 – “… Italian physician Luigi Galvani (1937-1798), who …”. I think, the years are not correct. Please correct.

Answer:Thank you very much for highlighting this incongruity. Galvani actually lived between 19737 and 1798. We have corrected this typo

  1. Introduction chapter well describes fundamental information about galvanism and study presented in the paper. References are correct.

Answer:No comments. Thank you

  1. Is Figure 1 made by authors or taken from references. If from references, add information what reference it is.

Answer:Figure 1 and all others are original and made by the authors.

  1. Subchapter 2.1. Materials. I think, it would be good to present some technical data of used measurement device, such as minimum of voltage, resolution, and others. Please complete if it is possible.

Answer:I attach a table with the main characteristics of the device. (word document)

We believe it would be more interesting to insert the internet address to consult all the characteristics and save space in the text. ”http://www.ictsl.net/productos/aparatos/phmetroportatilhi9024yhi9025hanna.html”

We regret that they are in Spanish.

Thank you for your contribution.

  1. Table 3 – please use the units of properties, you present in the table.

Answer:We have changed the tables as also suggested by referee 1.

Thank you for your time and effort to improve our work.

Reviewer 4 Report

The manuscript "Electrical Potentiometry with Intraoral Applications" is a very interesting scientific work that aimed to develop a system for measuring galvanic currents that would allow us to perform accurate measurements in the oral cavity and also sought to compare our new device with a reference electrode. This work is of great interest in the dental practice. The conducted methods are original and the manuscript is written well, but I have some suggestions to make this work clearer for readers. 

  1. Kindly, At the end of the abstract section, add your work conclusion and specific recommendations.
  1. Introduction: In the first paragraph, the citation is missing.
  1. Avoid the use of "we or our" throughout the whole manuscript.
  1. Add more updated references from 2020-2022 in the introduction and throughout the whole manuscript.
  1. Table 1 is not clear and needs some modifications.
  1. At the end of the material and method section, add a separate subsection—statistical analysis.
  1. At the end of the manuscript, add a conclusion part showing the novelty, outcomes, and recommendations.

  1. I suggest that this manuscript needs to be revised by a native speaker

Author Response

Referee 4

Comments and Suggestions for Authors

The manuscript "Electrical Potentiometry with Intraoral Applications" is a very interesting scientific work that aimed to develop a system for measuring galvanic currents that would allow us to perform accurate measurements in the oral cavity and also sought to compare our new device with a reference electrode. This work is of great interest in the dental practice. The conducted methods are original and the manuscript is written well, but I have some suggestions to make this work clearer for readers. 

  1. Kindly,At the end of the abstract section, add your work conclusion and specific recommendations.

Answer:We have added our conclusions at the end of the abstract:

“As a conclusion our system is able to detect galvanic currents in vitro and is we consider that it is capable of being employed in vivo so we recommend to determine the effects they may cause on dental implants.”

  1. Introduction: In the first paragraph, the citation is missing.

Answer:We have included the reference to the biography of Luigi Galvani.

“[1] “ 1.           Gillispie, C. Complete Dictionary of Scientific Biography; Charles Scribner’s Sons,: Detroit (Mich.) :, 2008; Vol. 9; ISBN 9780684315591.

  1. Avoid the use of "we or our" throughout the whole manuscript.

Answer:The use of we or our has been removed from the full text.

  1. Add more updated references from 2020-2022 in the introduction and throughout the whole manuscript.

Answer:We have included new references, but have kept most of the old ones, as these are the ones on which the work is based.

New references

  1. Gillispie, C. Complete Dictionary of Scientific Biography; Charles Scribner’s Sons,: Detroit (Mich.) :, 2008; Vol. 9; ISBN 9780684315591.

  • Soares FMS, Elias CN, Monteiro ES, Coimbra MER, Santana AIC. Galvanic Corrosion of Ti Dental Implants Coupled to CoCrMo Prosthetic Component. Int J Biomater. 2021 Oct 22;2021:1313343. doi: 10.1155/2021/1313343. eCollection 2021. PMID: 34721582 Free PMC article.
  • Stich T, Alagboso F, Křenek T, Kovářík T, Alt V, Docheva D. Implant-bone-interface: Reviewing the impact of titanium surface modifications on osteogenic processes in vitro and in vivo. Bioeng Transl Med. 2021 Jul 12;7(1): e10239. doi: 10.1002/btm2.10239. PMID: 35079626; PMCID: PMC8780039.
  • Lozano P, Peña M, Herrero-Climent M, Rios-Santos JV, Rios-Carrasco B, Brizuela A, Gil J. Corrosion Behavior of Titanium Dental Implants with Implantoplasty. Materials (Basel). 2022 Feb 19;15(4):1563. doi: 10.3390/ma15041563. PMID: 35208101 Free PMC article.
  • Verdeguer P, Gil J, Punset M, Manero JM, Nart J, Vilarrasa J, Ruperez E. Citric Acid in the Passivation of Titanium Dental Implants: Corrosion Resistance and Bactericide Behavior. Materials (Basel). 2022 Jan 12;15(2):545. doi: 10.3390/ma15020545. PMID: 35057263 Free PMC article.
  • Gürbüz-Urvasızoğlu G, Ataol M, Özgeriş FB. Trace elements released from dental implants with periimplantitis: a cohort study. Ir J Med Sci. 2022 May 7. doi: 10.1007/s11845-022-03020-y. Online ahead of print. PMID: 35524031
  • Nagay BE, Cordeiro JM, Barao VAR. Insight Into Corrosion of Dental Implants: From Biochemical Mechanisms to Designing Corrosion-Resistant Materials. Curr Oral Health Rep. 2022;9(2):7-21. doi: 10.1007/s40496-022-00306-z. Epub 2022 Jan 29. PMID: 35127334 Free PMC article. Review

  1. Table 1 is not clear and needs some modifications.At the end of the material and method section, add a separate subsection—statistical analysis.

Answer:We have changed the tables as also suggested by referee 1 and 3

We have included the statistical analysis section (page 5, line 121) .

  1. At the end of the manuscript, add a conclusion part showing the novelty, outcomes, and recommendations.

Answer:We have introduced the following sentence: Page 10, lines 253-255 “In conclusion, this system is capable of measurements in the 250Mvolt range and can be used in vivo on patients with implants. This will allow to study the possible effect of corrosion on bone loss around implants.”

  1. I suggest that this manuscript needs to be revised by a native speaker

Answer:Thank you for your recommendation. The text was reviewed by the American Journal Expert, whose certificate is attached. However, it has been re-edited by native speakers competent in the subject.

Thank you for the time and effort spent on improving this work.

Round 2

Reviewer 1 Report

It can be accepted in present form.

Reviewer 4 Report

The authors established well all the required corrections Thank you